# Swarm: A federated cloud framework for large-scale variant analysis

**Amir Bahmani** [1,2,3], **Kyle Ferriter** [2,3], **Vandhana Krishnan** [2,3], **Arash Alavi** [2,3], **Amir Alavi** [2,3], **Philip S. Tsao** [4,5], **Michael P. Snyder** [1,2,3]*, **Cuiping Pan** [5]*

**1** Stanford Healthcare Innovation Lab, Stanford University, California, United States of America, **2** Stanford Center for Genomics and Personalized Medicine, Stanford University, California, United States of America, **3** Department of Genetics, Stanford University, California, United States of America, **4** Division of Cardiovascular Medicine, Stanford University, California, United States of America, **5** Palo Alto Epidemiology Research and Information Center for Genomics, VA Palo Alto, California, United States of America

☉ These authors contributed equally to this work.
* mpsnyder@stanford.edu (MPS); cuiping@stanford.edu (CP)

## Abstract

Genomic data analysis across multiple cloud platforms is an ongoing challenge, especially when large amounts of data are involved. Here, we present Swarm, a framework for federated computation that promotes minimal data motion and facilitates crosstalk between genomic datasets stored on various cloud platforms. We demonstrate its utility via common inquiries of genomic variants across BigQuery in the Google Cloud Platform (GCP), Athena in the Amazon Web Services (AWS), Apache Presto and MySQL. Compared to single-cloud platforms, the Swarm framework significantly reduced computational costs, run-time delays and risks of security breach and privacy violation.

## Author summary

With more and more genomic data generated and stored in different computational platforms, federated computation has become an area of strong interest. Our software framework Swarm provides such a solution. With Swarm, large genomic datasets hosted on different cloud platforms or on-premise systems can be jointly analyzed with reduced data motion. It not only enables more economical computation, but also enables collaboration within or between different organizations and institutions, empowering multi-cloud solutions as long as the required users' authorization and credentials are available. Moreover, it accelerates discoveries in both inter- and intra- organizations. For instance, information on disease variants from healthcare data can be quickly shared between two hospitals potentially seeking faster treatments if procurable. We demonstrated the utility of Swarm in different application cases, including variant query, functional annotation information query, their joint computation between databases hosted on different platforms, and a potential application in federated learning across cloud platforms.

**Data Availability Statement:** All relevant data are within the manuscript and its Supporting Information files.

**Funding:** The computational and storage cost of this work, and CP,VK,PST were supported by the

Veterans Affairs Office of Research and Development Cooperative Studies Program (https://www.research.va.gov/default.cfm). MPS, AB, KF, ARA, AMA received support by National Human Genome Research Institute at the United States National Institutes of Health (U24 HG009397 356 and RM1-HG007735), and by the generosity of Eric and Wendy Schmidt by recommendation of the Schmidt Futures program (https://schmidtfutures.com/). The content is solely the responsibility of the authors and does not necessarily represent the official views of the VA Healthcare System, the National Human Genome Research Institute, the National Institutes of Health, or the Schmidt Futures program. The funders had no role in design, data processing, implementation, decision to publish, or preparation of the manuscript.

**Competing interests:** I have read the journal's policy and the authors of this manuscript have the following competing interests: MPS is the Cofounder and SAB member of Personalis, Mirvie, SensOmics, Qbio, January, Oralome, Filtricine, Protos; SAB of Genapsys, Jupiter.

This is a *PLOS Computational Biology* Software paper.

## Introduction

Big data is radically transforming precision medicine and the landscape of information technology for the life sciences. For instance, the exponential growth of data in genomics research is shifting the data generation bottleneck from sequencing costs to new storage and computation needs [1]. Scalable and distributed computing is integral to solving these bottlenecks. Cloud computing offers elastic scalability and a flexible pay-as-you-go model that removes complex maintenance workloads from end users, equips researchers to store massive data in addition to performing intensive computing [2], and enhances data sharing and collaboration; therefore, it is well suited for large-scale genomic analysis [3–5]. Currently, multiple cloud service platforms are in use by the genomic community for data hosting, computing, and sharing. For example, Encyclopedia of DNA Elements [6] mainly employs Amazon Web Services (AWS), while the Genome Aggregation Database [7] data is stored on Google Cloud Platform (GCP). However, due to the platform heterogeneity, joint analyses across cloud platforms are not optimized. Indeed, data motion across cloud platforms is non-trivial, as the amount of data involved is often large and sometimes even unnecessary, resulting not only in high transfer costs, but also drastic delays and a plethora of security and privacy risks.

Here, we propose a new framework, Swarm, for federated computation among multiple cloud platforms and on-premise machines. With Swarm, we developed an API that leverages a serverless computing model [8] to evaluate data motion needs, perform computation *in situ* as much as possible, and facilitate data transfer if moving data between different platforms becomes necessary. We tested the utility of our framework in several research scenarios involving genomic variants and showed an optimized data crosstalk solution where the cost associated with data motion was drastically reduced. This federated computational framework will enable researchers under the same security protocols and data access rights to freely utilize services provided by different computational platforms without being locked into any specific features of the original data hosting platform. Performing minimum data motion between cloud providers in the Swarm federated computational framework also improves the security and privacy of health data being transferred. Various cloud providers encapsulate best practices to support health privacy regulations such as the Health Insurance Portability and Accountability Act (HIPAA)—e.g., HIPAA Compliance on GCP [9] and AWS [10]. Leveraging these logging, auditing, and monitoring best practices and using strong authentication and encryption mechanisms for data in transit would help reduce the security and privacy risks associated with multi-cloud orchestration.

## Design and implementation

### Overview of Swarm

Fig 1 shows the architecture of Swarm for federated computation on genomic variants. Swarm classifies variant inquiry tasks into two main categories. "Stat Query" handles all queries that do not require data motion, and returns statistics such as counts of matched records and frequency of the alleles. "Data Query" handles queries that involve moving a set of records to another computing platform for further processing. To handle heterogeneity in how the data are formatted, we adopted the *BED* file format: chromosome_name, start_position, end_position, reference_bases, and alternate_bases, while the remaining fields are left unchanged to accommodate more specific information. Specifically, to minimize data motion, Swarm first

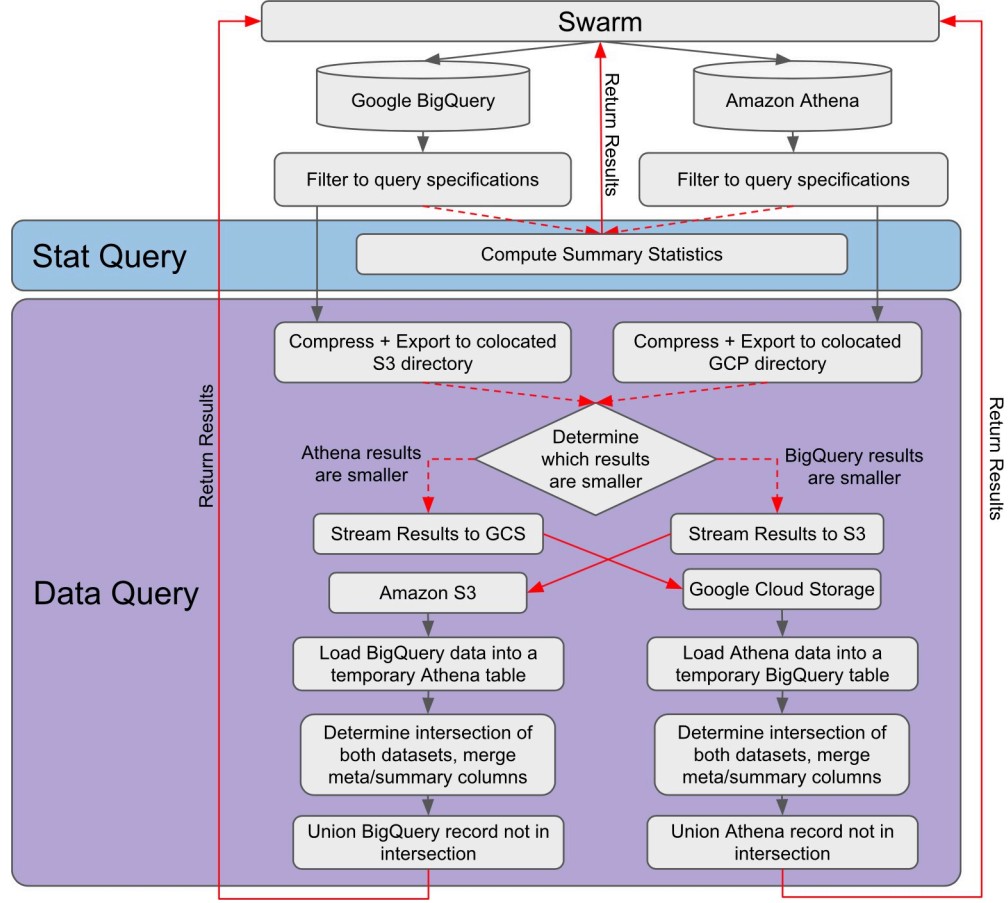

**Fig 1. Swarm Framework: The Swarm architecture enables federated computation on genomic variants.** It classifies variant inquiry tasks into two main categories. "Stat Query" handles all queries that do not require data motion, and returns statistics such as counts of matched records and frequency of the alleles. "Data Query" handles queries that involve moving a set of records to another computing platform for further processing. In this figure, as an example, we illustrate the use of AWS Athena and GCP BigQuery.

evaluates the amount of data returned from each platform, selects and compresses the smaller set, and then moves it to the other computing platform where the larger dataset is hosted. After transforming the small dataset to a temporary table in the new computing environment, Swarm joins the two datasets and returns the final result to users. In addition to the default option of moving the smaller dataset, Swarm also provides an option to move the larger data, for example to better comply with the privacy and security requirements of users' projects.

## Implementation of Swarm

Our current implementation utilized two commercial serverless and interactive query services: 1) Google BigQuery: https://cloud.google.com/bigquery/ from *GCP* and 2) Amazon Athena: https://aws.amazon.com/athena/ from *AWS*. Swarm also supports Apache Presto, an open source distributed query engine that supports much of the *SQL* analytics workload at Facebook [11]. All the above platforms support the *SQL* standard, which enables a high degree of portability and interoperability [12]. The proposed framework can be extended to other computational platforms following a similar achitecture.

## Configuring Swarm framework for the use cases

1. For our first experiment, we randomly split the 1000 Genomes Phase 3 VCF in half and stored one half in BigQuery and the other half in Athena. Swarm queried the four variants shown in Table 1 on the split datasets and computed their corresponding allele frequencies. Such data inquiries do not involve data motion and are categorized as "Stat Query". The initial set of configurations did not include any optimizations (e.g., partitioning or clustering), while in the second set of configurations, we split these tables into 4,000 partitions over the start positions for the BigQuery and Athena tables. Additionally, the BigQuery table was clustered by reference name and start position. Each Athena partition was created as one Apache Parquet [13] data file.

2. In the second experiment, we examined data inquiries that involved data motion termed "Data Query". These types of queries involved joining datasets from multiple cloud platforms for performing the combined analysis. For this, we studied a case of variant annotation in which the variant file containing the 1000 Genomes calls was stored in an Athena table in AWS, whereas the reference files for functional annotation of the genotypes were collectively stored in a BigQuery table on GCP. We partitioned both of these tables into 4,000 partitions based on the start position. For the BigQuery annotation reference table, we also applied clustering using both the start position and reference name.

3. The third experiment was run on Apache Presto (12.4. Release 331) configured on Google DataProc (v1.5). As opposed to the serverless model of the BigQuery and Athena experiments where charges were based on the amount of data traversed, the Apache Presto experiment was run on a DataProc cluster with pre-allocated computing resources and charges were based on the time that storage, memory, and CPU resources remain allocated. In this experiment, we loaded the second half of the 1000 Genomes dataset that was used in the first experiment for Amazon Athena. This table comprised 4,000 parquet files partitioned based on their start positions in Amazon Athena. We created DataProc clusters with different numbers of worker nodes (all n1-standard-4 machine types). Although we configured Apache Presto on Google Cloud Dataproc, note that it can also be configured on any HPC clusters.

4. The fourth experiment was run on *MySQL 5.7* as a traditional relational database system (RDBMS). *MySQL 5.7* is fully multi-threaded, and makes use of all vCPUs made available to it. For this experiment, we loaded the entire 1000 Genomes dataset without the genotyping columns. The *MySQL* queries were executed on different n1-standard machine types on Google Cloud Platform. In order to compare the performance of Apache Presto on a CSV file and a Parquet file based on rsID search, the same 1000 Genomes dataset without genotyping information was utilized. Similar to the third experiment, the runs on Apache Presto were performed using a different number of n1-standard machine type-based worker nodes on Google Cloud Platform.

**Table 1. Variants used for testing Stat Queries.**

| Description | rsID | Chr | Pos |
|---|---|---|---|
| Attention-deficit/hyperactivity disorder (ADHD) | rs671 | 12 | 112241766 |
| Blue Eye Color (BEC) | rs12913832 | 15 | 28365618 |
| Coronary Heart Disease (CHD) | rs1333049 | 9 | 22125503 |
| Lactose Intolerance | rs4988235 | 2 | 136608646 |

5. As a proof of concept, we also implemented a version of Swarm that supports containerization for *ad hoc* computation such as federated machine learning tasks. Users can provide an image for one platform (e.g., training a model), and Swarm transfers the output model/files to the other platform and continues the computation by creating a new container on the second platform. For this experiment, we followed the tutorial on https://choishingwan.github.io/PRS-Tutorial/ [14] to build a polygenic risk score (PRS) model from a height genome-wide association study, transferred the model via Swarm to another cloud, and applied it to the 1000 Genomes dataset.

### Statistical tests

All our experiments were run four times. Average values and standard deviations were computed. For comparing differences between a pair of experiments, F tests were used to evaluate variances and two sample t-tests were used for computing P values. For comparing within a group of experiments, e.g., runtimes with varying number of nodes in the Apache Presto runs, anova tests were used for computing P values.

## Results

We demonstrated the utility of Swarm in facilitating genomic variant analysis across datasets on different cloud platforms, such as BigQuery on GCP, Athena on AWS, and Apache Presto. While BigQuery and Athena are serviced columnar databases, Apache Presto runs on Hadoop clusters. Here we used Dataproc, a managed Apache Hadoop [15] and Apache Spark [16] service with pre-installed open source data tools for batch processing, querying, streaming, and machine learning.

A common use case is to query particular variants in a dataset. When genomic positions and genotypes are provided, our API can query datasets on BigQuery and Athena to retrieve records of the matched variants. Similarly, if the HUGO Gene Nomenclature Committee (HGNC) [17] symbol of a gene or the positions of a genomic region are given, our API can return variant records that fall into that region. It can also perform computation based on standard SQL, such as computing allele frequency for a specific variant. We conducted the following experiments using Swarm, each mapping to the use cases described in the previous section: Experiment 1: computing allele frequencies for datasets across two cloud platforms, Experiment 2: annotating an input gene using functional reference datasets from another cloud, Experiment 3 and Experiment 4: retrieving records of a variant given the rsID (i.e., the matched VCF fields) in databases of different structures, and Experiment 5: a proof-of-concept study of transferring a genomic model from one cloud to another cloud.

### Datasets

To evaluate the capability of Swarm, the 1000 Genomes Project's variant calling results [18] were used: https://cloud.google.com/life-sciences/docs/resources/public-datasets/1000-genomes. Variants indicated in Table 1 were queried in the 1000 Genomes dataset via Swarm. Annotation of the 1000 Genomes dataset was performed using the resources listed in Table 2.

### Stat Query: Allele frequency computation across clouds

We tested our system, Swarm, using the first experimental set-up as described in Design and Implementation. Briefly, half of the 1000 Genomes samples and their genomic variants were stored in GCP and the other half were stored in AWS, and we used swarm to compute variant

**Table 2. Databases used for annotating the 1000 Genomes data sets in this study.**

| Annotation Dataset Name | Reference |
| --- | --- |
| dbNSFP 35a | https://sites.google.com/site/jpopgen/dbNSFP |
| 1000Genomes | https://www.internationalgenome.org/data/ |
| RegulomeDB | https://www.regulomedb.org/regulome-search/ |
| ClinVar | https://ftp.ncbi.nlm.nih.gov/pub/clinvar/vcf\_GRCh37/ |
| UCSC gtex Eqtl Cluster | https://genome.ucsc.edu/gtex.html |
| gnomAD | https://gnomad.broadinstitute.org/downloads |
| Cosmic70 | https://annovar.openbioinformatics.org/en/latest/user-guide/filter/#cosmic-annotations |
| Dann | https://annovar.openbioinformatics.org/en/latest/user-guide/filter/#dann-annotations |
| Eigen | https://annovar.openbioinformatics.org/en/latest/user-guide/filter/#eigen-score-annotations |
| eQTL & Diseases | http://www.exsnp.org/Download |
| GTEx Analysis v7 eQTL | https://gtexportal.org/home/datasets |
| SNP151 | https://genome.ucsc.edu/cgi-bin/hgTrackUi?db=hg38\&g=snp151 |
| Wellderly | https://genomics.scripps.edu/browser/files/wellderly/vcf |

allele frequencies in each cloud, and subsequently merged the results for deriving the combined allele frequencies. To optimize the performance, we introduced the methods of partitioning and clustering. Fig 2A depicts the execution time with or without optimizations. Fig 2B shows the amount of data processed.

For the genotypes in Table 1, partitioning and clustering significantly improved runtimes. Swarm processed 0.15GB data from BigQuery for calculating their allele frequencies in one half of the samples in the 1000 Genomes dataset. Given that the whole BigQuery table was 540GB, this was a small fraction. Compared to the scenario of moving the dataset to the second cloud, Swarm computed *in situ* and reduced the egress cost by 99.98%, and in this particular example, users could run approximately 3,600 similar queries before reaching the break-even point. Note, this experiment used only half of the 2,500 samples from the 1000 Genomes Project. Larger cohorts with hundreds of thousands or millions of samples can benefit largely from systems like Swarm.

## Data Query: Comprehensive functional annotation of variants or genes across clouds

In the second experiment, we mimicked a scenario where the variant calling results were stored in one computing platform (e.g., AWS Athena), and the reference files for functional annotation (Table 2) were stored in another (e.g. GCP BigQuery), and users wanted to annotate all variants pertaining to the *TP53* gene. Our API determined that the smallest dataset to transfer would be the extracted entries of the *TP53* functional annotation records from BigQuery. This subset of records from BigQuery containing the *TP53* functional information was compressed and sent in an encrypted format to AWS, stored in a temporary AWS Athena table, and then used to perform joint analysis with the functional reference table.

Fig 3 shows the performance of Swarm for annotating a set of genes. In this case, Swarm moved the annotation records that overlap with the input genes from BigQuery to Athena and then performed annotation. Again, our optimization using partitioning and clustering significantly shortened the runtime and reduced the amount of data scanned. Here, Swarm processed 99.4 MB in BigQuery and 13.5 MB in Athena, and then compressed and copied a 3.1

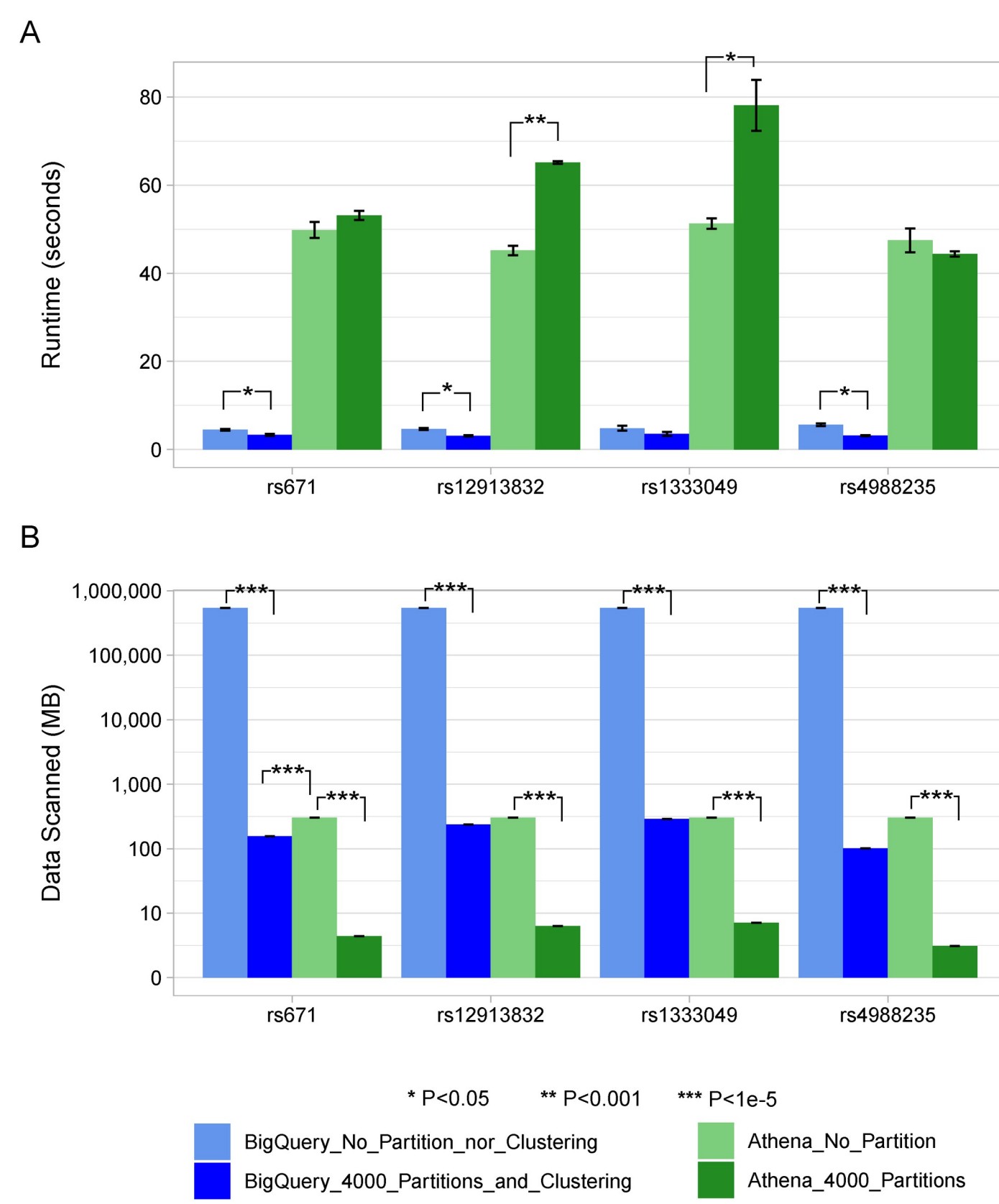

**Fig 2. Runtime and amount of data processed for computing allele frequency for an input set of rsIDs in BigQuery and Athena.** Average values and standard deviations were plotted. (**A**) depicts the average execution time in seconds. The light blue and light green bars represent configurations without any optimizations (i.e., the entire input data used as it was), and the dark blue and dark green bars represent configurations with optimizations (i.e., the input data was divided by partitioning or clustering); (**B**) shows the amount of data processed in megabytes, and the y-axis is logarithmic in scale. Significance differences between groups are indicated on top of the bars (two samples t-test). Note that for each rsID experiment, differences in runtimes between any BigQuery and Athena runs in (A) were highly significant (P < 1e-5), and for (B), differences within the BigQuery or Athena runs were also highly significant (P < 1e-5).

MB subset of the 233 GB annotation table from BigQuery to Athena, in contrast to copying the whole annotation table to the other side of the computation in Athena. This resulted in moving only 0.001% of the full annotation table.

## Variant query in Apache Presto

With the third computing configuration set-up as described in Design and Implementation, we demonstrated the cloud-agnostic feature of the Swarm platform. This was achieved by assessing the query processing times using the rsID *rs671*, provided in Table 1 to compute allele frequencies. Table 3 shows the average runtime for querying *rs671* for the cluster with different numbers of nodes.

The short computation time stems from the fact that these sets of nodes were dedicated to Presto experiments (i.e., after the process of partition discovery, these nodes cache metadata). On the other hand, in the serverless systems (e.g., BigQuery and Athena), for every input query, a new set of nodes unfamiliar with the structure of the data are prospectively assigned, increasing the query response times in the absence of caching. Our Presto cluster had a significantly faster execution time than Amazon Athena. Note that the serverless system leverages auto-scaling features and scales its clusters automatically.

The runtimes for searching for *rs671* in a Dataproc cluster using partitioning versus ignoring partitioning are displayed in Fig 4A. Also, average runtime using partitioning with preemptible and non-preemptible instances is shown in Fig 4B. In this example, for the preemptible cluster among *N* worker nodes, only two were non-preemptible instances (Non-PVM); the rest were preemptible (i.e., N-2 PVMs). The execution time was almost identical in the above tested example. However, Fig 4C shows a large cost reduction for using preemptible instances as the number of worker nodes increased. Dataproc uses preemptible instances as secondary workers to scale computation without scaling storage. This is because preemptible instances are not suitable for HDFS storage, since a preemption would impact the availability of HDFS blocks.

Note that in this third example, because Presto is a columnar database, every VCF field in the 1000 genomes VCF file was stored in a columnar format. Therefore, for the input Stat Query on an rsID, Presto scanned the column associated with the corresponding rsID field. The speed-up, displayed in Fig 4A deteriorated as the number of nodes increased, indicating that there was not enough work for each worker node, and a significant portion of overall turnaround time was spent on sequential processing, rather than parallel worker processing.

## Variant query in MySQL

With respect to interoperability, we also demonstrated the utility of Swarm in facilitating genomic variant analysis on *MySQL*. Fig 5A depicts the performance of *MySQL 5.7* for searching *rs671*. For this experiment, *MySQL* performs well on the instance with 8 vCPUs and 30 GB memory, which is mainly due to the table size. The size of the VCF table is 15.97 GB, so the table fits well into the main memory of the instance.

Fig 5B depicts the performance of running a query on a table imported as CSV as opposed to the Parquet version, on Apache Presto. As shown, the amount of data scanned for Parquet

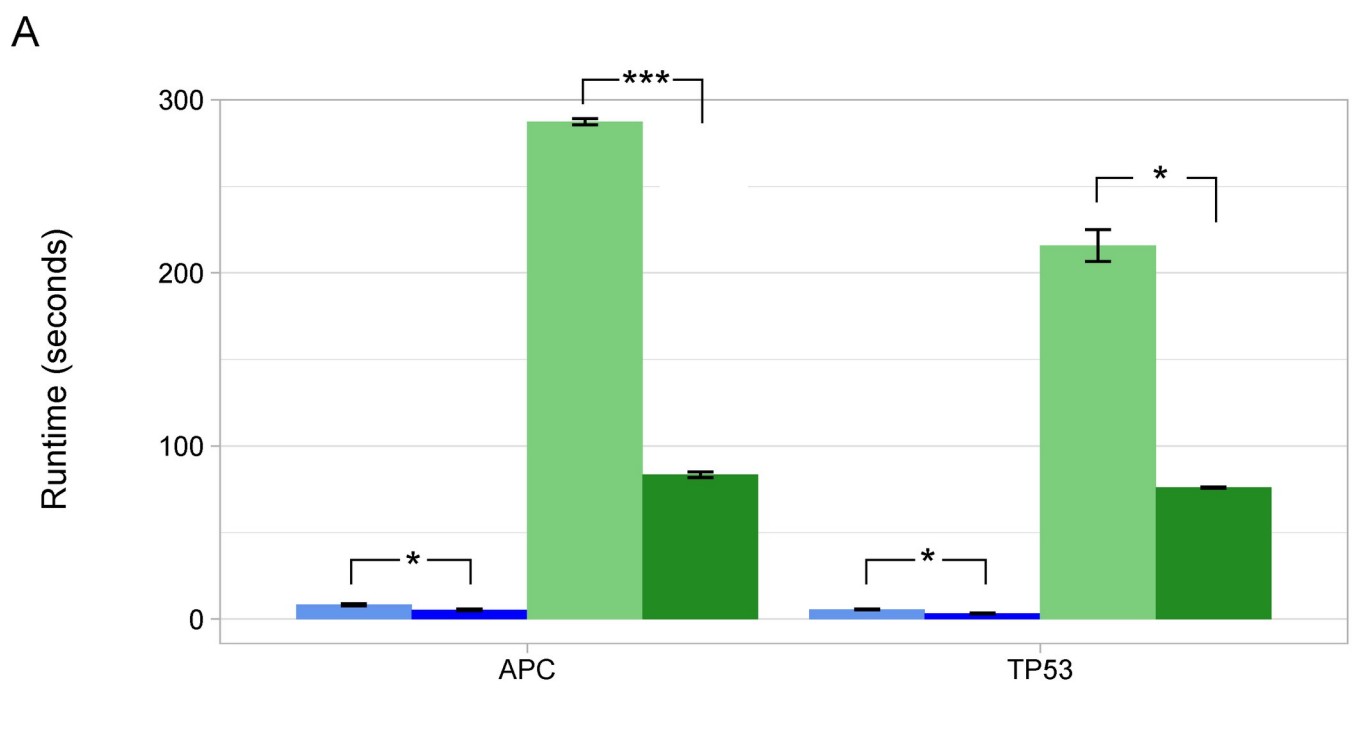

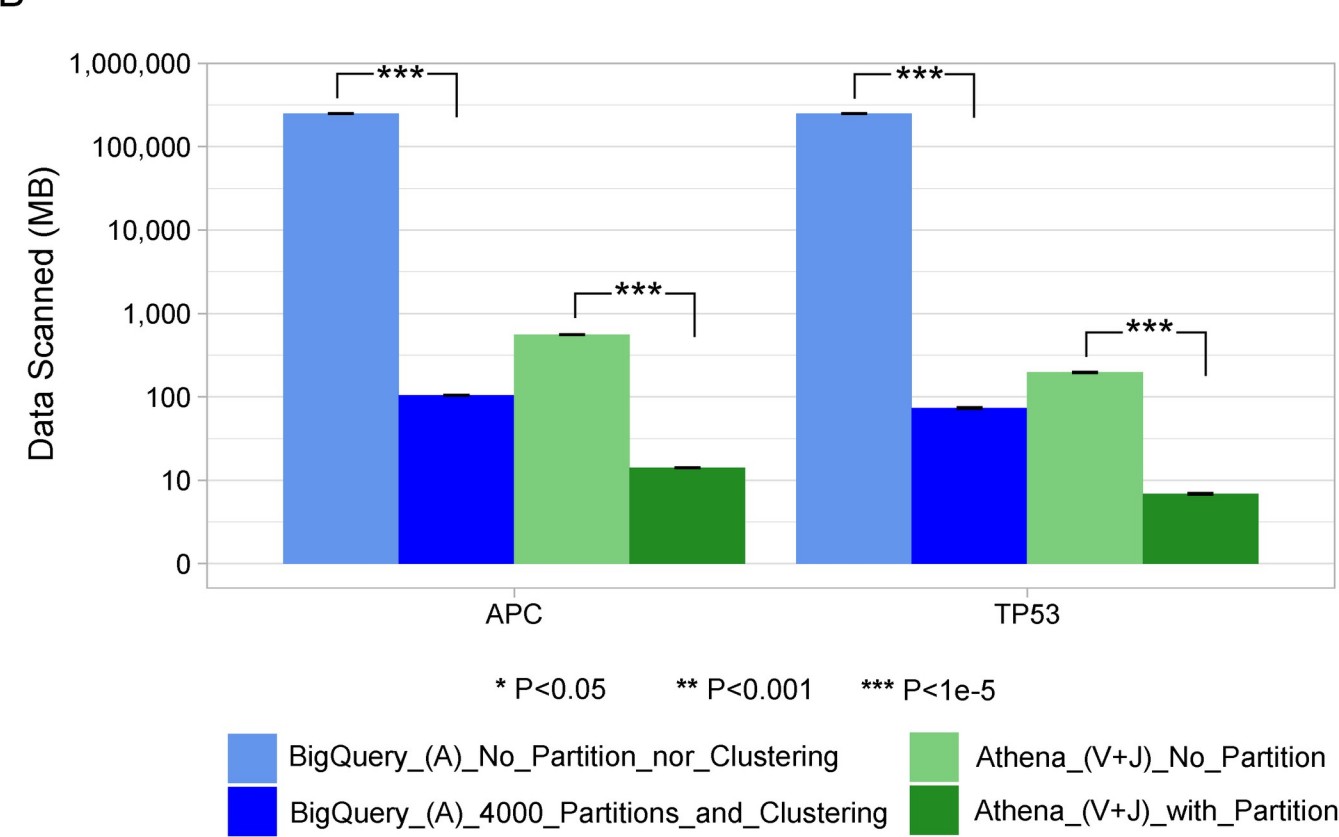

**Fig 3. Runtime and the amount of data processed for annotating an input set of genes.** A, V and J stand for Annotation records, Variant records and Join table operations, respectively. Average values and standard deviations were plotted. (**A**) depicts the execution time in seconds for the two input genes. In this experiment, the annotation table was in BigQuery and the variant table in Athena. Therefore, Swarm first found all the annotation records in BigQuery that

overlapped with the input gene regions, compressed them and moved them to Athena. Then, on the Athena side, Swarm decompressed the overlapping annotation data and created a temporary table, which was eventually processed to join with the existing variant table. The light blue and light green represent the configurations without any optimizations by partitioning or clustering, and the dark blue and dark green represent the configurations with optimizations. (**B**) shows the amount of data processed in megabytes, and the y-axis is logarithmic in scale. Significance differences between groups are indicated on top of the bars (two samples t-test). Note that for (A), differences between any BigQuery and Athena groups were highly significant (P < 1e-5), and for (B), differences within the BigQuery or Athena groups were also highly significant (P < 1e-5).

was 1.27 GB, which was significantly smaller compared to the CSV version of 12.7 GB. This shows that Parquet as a columnar storage format, is efficient for columnar distributed SQL query engines such as Apache Presto.

For both experiments in Fig 5, we used the 1000 Genomes VCF by leaving out the genotyping columns. The *MySQL* experiment illustrates vertical scaling (i.e., increasing the number of vCPUs per worker node) whereas the Presto experiment represents horizontal scaling (i.e., increasing the number of worker nodes). As a simple comparison, it can be observed that *MySQL* run on a worker node with 8 vCPUs (CSV: 63.2625 seconds) is significantly slower than the Apache Presto run on two worker nodes with collectively 8 vCPUs (Parquet: 12.205 seconds, CSV: 29.005 seconds).

## An example of federated learning

In addition to querying variants and computing summary statistics, Swarm can in principle facilitate federated learning by transferring models across clouds. We demonstrated this potentiality via the computation of genomic polygenic risk scores (PRS) (see Design and Implementations). We first learned the PRS model based on the summary statistics of a genome-wide association study (GWAS) on height, and then applied the model to the European sub-population in the 1000 Genomes dataset, which were stored in a different cloud platform, for constructing the PRS for individual genomes. This represents a common scenario where large computing can be performed in one cloud platform for deriving a statistical model, and the actual application of the model is elsewhere. In addition, Swarm can help transfer intermediate results of the machine learning models across the cloud, so that the model can continue to learn and improve using the new data in the second cloud. For instance, gradients of deep learning models can be transferred by Swarm. However, we note that currently model transfer in Swarm uses localized sever, and further development in security measures are necessary. Particularly, individual genomes are regarded as protected data in healthcare, so model transfer should use caution. In the Future Directions, we elaborate on a few promising techniques for strengthening the security feature for federated learning.

## Availability

Swarm is available as an open source tool at https://github.com/StanfordBioinformatics/Swarm

## Future directions

In this paper we presented Swarm, a framework for federated computation that promotes minimal data motion and facilitates crosstalk between genomic datasets stored on various cloud

**Table 3. Average execution time for querying *rs671* with the binID on the partitioned Parquet files of one half of the 1000 Genomes dataset using Apache Presto, with different numbers of worker nodes.** Each configuration additionally includes one master node.

| Number of Worker Nodes | 2 | 4 | 8 | 16 |
|---|---|---|---|---|
| Average Runtime (seconds) | 3.6725 | 3.1500 | 3.1900 | 3.9375 |

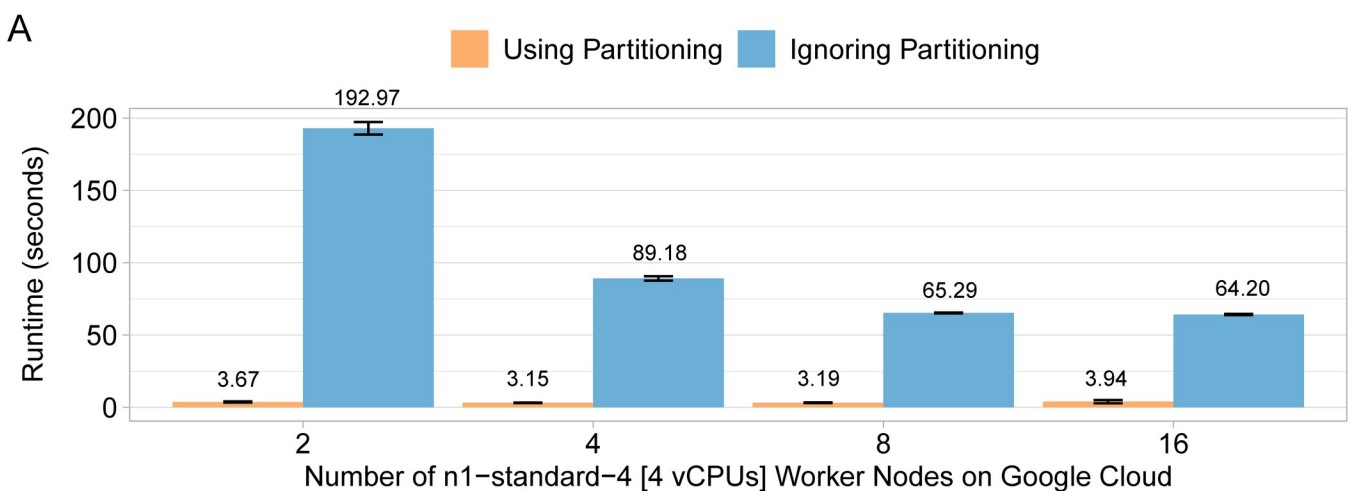

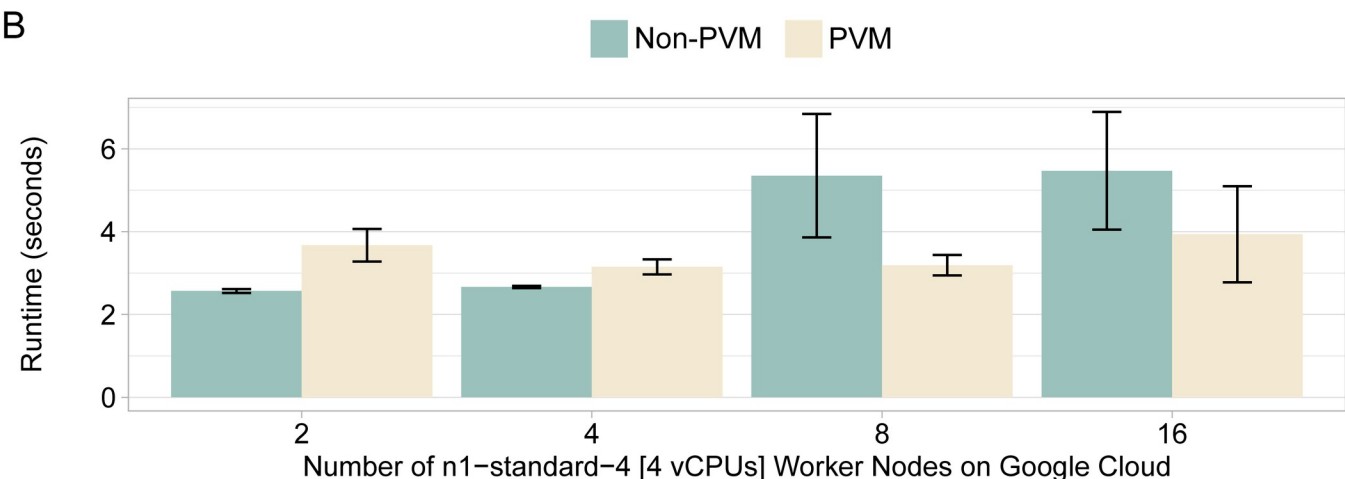

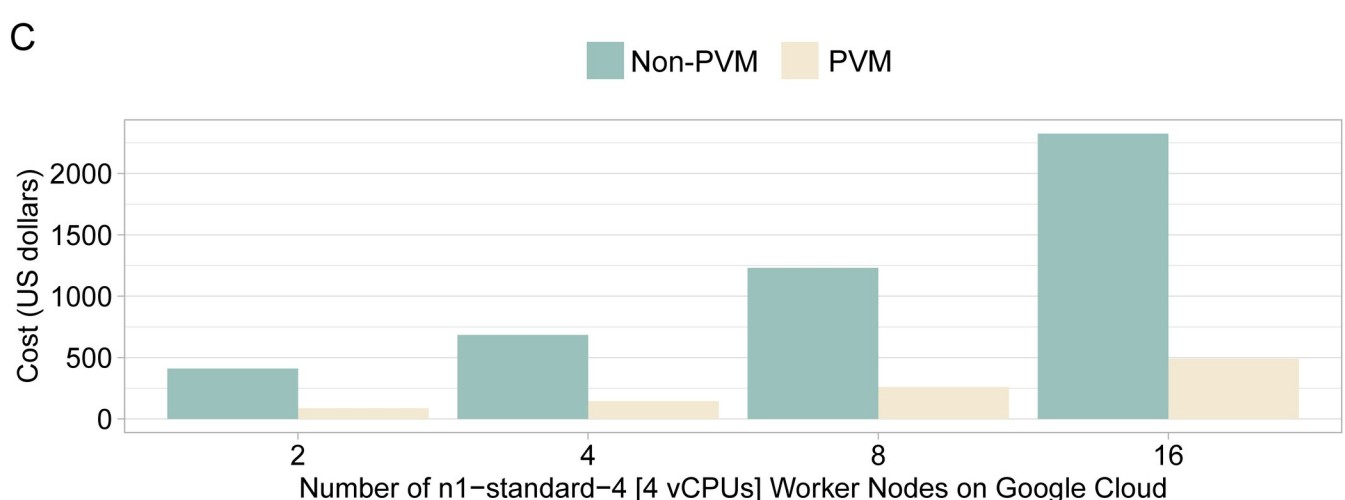

**Fig 4. Execution time for searching *rs671* with different number of worker nodes for running Apache Presto on Dataproc.** (**A**) The average runtime using partitioning versus ignoring partitioning in Apache Presto. (**B**) The average runtime using preemptible (PVM) and non-preemptible (Non-PVM) instances. Average values and standard deviations were plotted. (**C**) The projected cost of reserving the dedicated nodes on GCP on a monthly basis. Monthly cost as of February 2021 https://cloud.google.com/compute/all-pricing. Note, for serverless systems like BigQuery and Athena, users are charged based on the amount of data processed, respectively. In (A), Differences between the paired groups of with or without partitioning were highly significant (two sample t-tests P < 1e-5). In (B), differences between the paired groups of Non-PVM and PVM, although not significant, had marginal P values close to 0.05.

platforms. We demonstrated that it accurately performs genomic analysis across multiple databases on different cloud platforms, including GCP BigQuery, AWS Athena, Apache Presto on an Apache Hadoop cluster (e.g., Google Dataproc), and the traditional row-based database MySQL. For the tested computing platforms, we observed an obvious reduction in run-time and cost by utilizing partitioning and clustering techniques.

Our evaluations were scripted in Python against the REST interface provided by Swarm. Timing and data values were automatically extracted from output messages provided by the REST server. Each query was performed over four averaged runs and negligible variance was observed. The outputs from the queries spanning BigQuery and Athena were validated by comparing the original data files.

While Swarm performed well for the tested experiments, it is worth mentioning some of the limitations associated with the underlying computing frameworks. Currently in BigQuery, the maximum number of partitions per partitioned table is 4,000, and the maximum number of columns is 10,000 https://cloud.google.com/bigquery/quotas. At the same time, AWS Athena has the ability to support a maximum of 20,000 partitions per partitioned table https://docs.aws.amazon.com/athena/latest/ug/service-limits.html. Furthermore, BigQuery and Athena scale automatically based on the workloads. In contrast to Athena and BigQuery, Apache Presto utilizes Hadoop cluster, a different compute environment based on distributed computing architecture. We used Google Dataproc, a managed cloud service for running Apache Spark and Apache Hadoop. System engineers need to tune their clusters based on the number of requests and the size of the table, as well as the complexity of the queries. Although there are no explicit hard limits for the number of columns in Apache Presto, users are limited by available memory and maximum size of collection in Java. Google Dataproc used for Apache Presto has options to use either preemptible or non-preemptible secondary worker nodes, thereby helping reduce costs. Additionally, Google Dataproc used for Apache Presto required a minimum of two worker nodes.

In Google BigQuery, a feature for performing integer range partitioning into fixed-sized bins was added recently and was found to perform well under our experimental workloads. In Amazon Athena, performing integer range bin partitioning within Athena itself was not possible due to the Athena time and memory constraints on jobs. Since partitioning could not be performed within Athena, we instead performed this separately on an AWS EMR cluster using Python and Spark.

We showcased how the Swarm framework could accommodate federated machine learning tasks. Federated learning facilitates model training without the need of sharing raw data, and therefore strengthens privacy protection. However, it has been shown that federated learning is vulnerable to attacks against machine learning models such as model inversion, membership, or properties inference attacks [19–21].

This emphasizes the need for security and privacy preservation mechanisms in federated learning systems. A previous study showed that individuals in the beacon service of the Global Alliance for Genomic and Health (GA4GH) study were susceptible to re-identification attacks by allele-presence only queries [22]. Prohibiting anonymous access via monitoring and implementing best practices access control play the most important role for improving security and

A

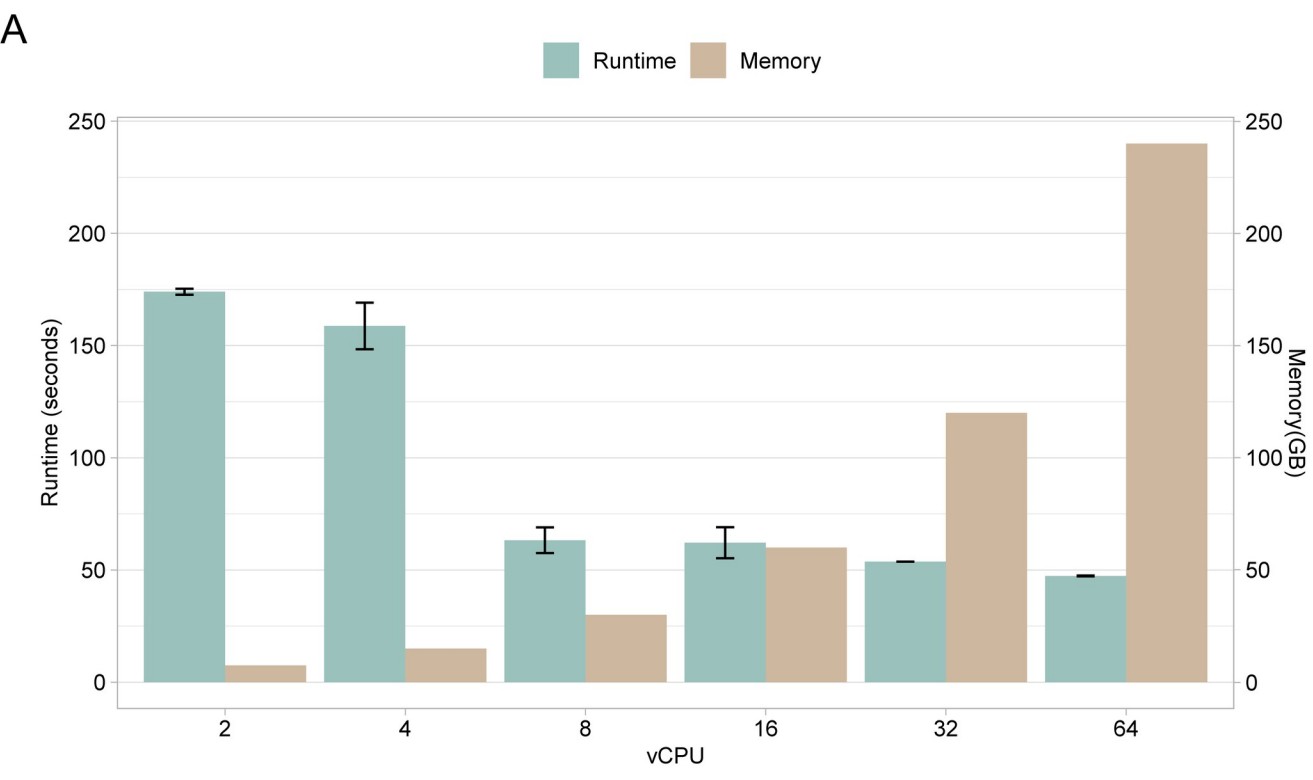

B

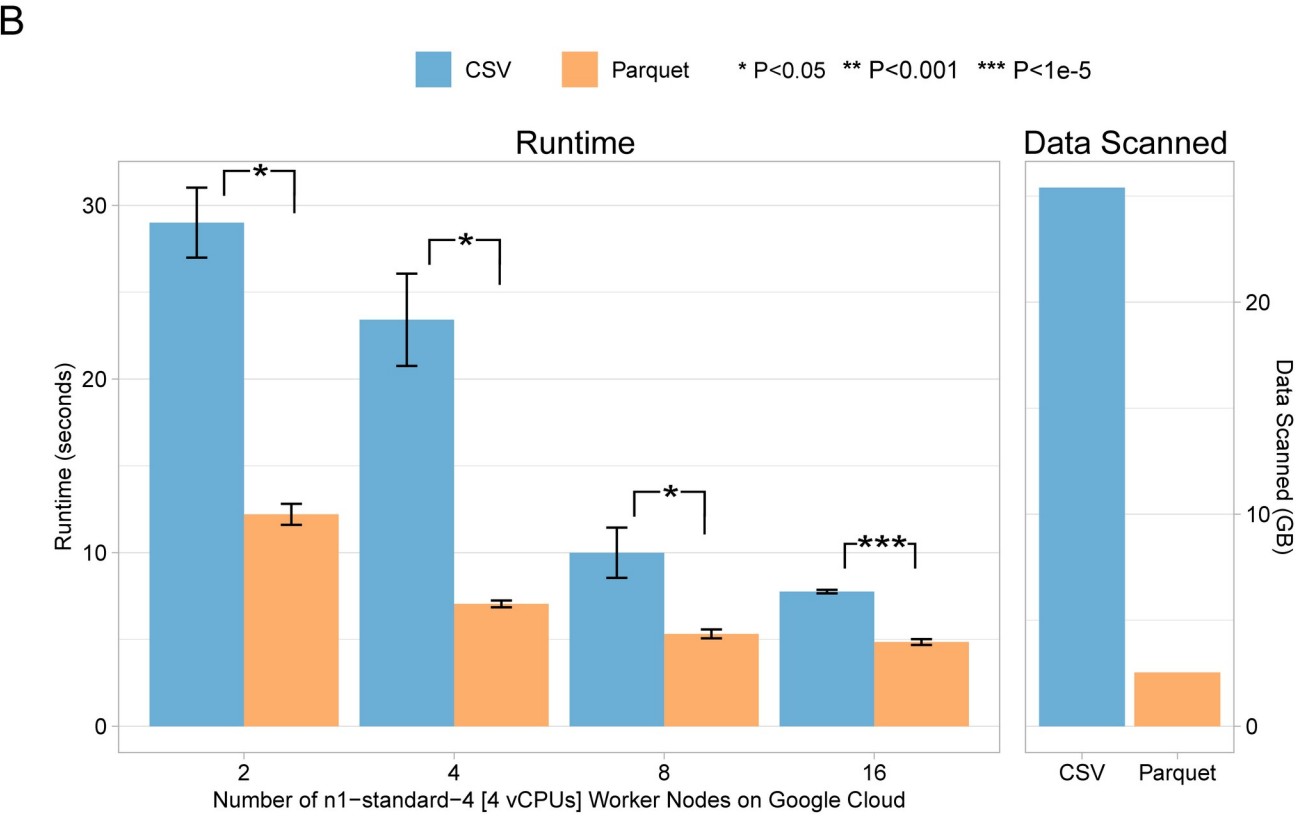

**Fig 5.** Searching *rs671* in the 1000 Genomes dataset loaded in (**A**) *MySQL* and (**B**) Apache Presto with different settings, i.e., varying number of vCPUs and main memory sizes. Average values and standard deviations were plotted. For the Apache Presto runs in (B), runtimes between CSV input versus Parquet input were compared, and significant P values are indicated (two sample t-tests). In addition, anova tests indicated that the number of worker nodes had a significant impact on the runtimes ($P < 1e-5$).

reducing loss of privacy. On the other hand, in the case of unauthorized accesses, minimizing data motion as enabled by Swarm can effectively protect data against data in-transit attacks (e.g., man-in-the-middle).

Further privacy protection techniques can be integrated into our proposed framework: 1) Secure Multiparty Computation (SMPC). SMPC cryptographic protocols allow multiple parties to jointly compute a function over their inputs while ensuring complete zero knowledge about the inputs. However, while the SMPC protocols preserve the privacy of inputs and processes among parties fairly, the privacy concerns about the output need to be addressed; 2) Differential Privacy. These methods add noises to the data in a way to simultaneously deliver high anonymization and utility preservation. The Swarm framework can be adapted to support differential privacy, K-anonymity, and K-isomorphism anonymization schemes to the data. 3) Homomorphic encryption. This is another cryptographic protocol that can protect data privacy through computing over encrypted data without access to the secret key. The main bottleneck of such computation-heavy privacy-preserving techniques is latency.

Moreover, for a secure cloud-to-cloud model transfer, one can use traditional asymmetric encryption methods (e.g., RSA) or leverage the built-in account-based transfer services (e.g., GCP Storage Transfer Service) to securely transfer models to the other cloud. Currently the GCP Transfer Service, AWS DataSync, and AzCopy support secure data transfer to/from Google cloud storage, Amazon S3 bucket, Microsoft Azure storage, or object URLs. For the object URLs, signed URLs can be used for limited permission and time access.

Swarm can also be extended to use in-memory transfer of data without using local storage for caching. Additional improvement for security and performance would be to leverage existing libraries for point-to-point transfer between cloud storage services without needing to pass through the Swarm server until the very end when data must be returned.

Another improvement could be in expanding the core code APIs of the service to make them more flexible and capable of accommodating data types beyond variants and variant annotation. This would enable more generic querying and adding functionality comparable to dataframes in the Pandas or Spark libraries, where columns can be detected and arbitrary table keys can be defined more broadly.

## Acknowledgments

We acknowledge the Stanford Genetics Bioinformatics Service Center (GBSC) for providing the gateway to GCP and AWS for this research. We thank members of the MVP bioinformatics team of Stanford University and VA Palo Alto for constructive feedback.

## Author Contributions

**Conceptualization:** Amir Bahmani, Kyle Ferriter, Vandhana Krishnan, Arash Alavi, Cuiping Pan.

**Data curation:** Amir Bahmani, Kyle Ferriter, Cuiping Pan.

**Formal analysis:** Amir Bahmani, Kyle Ferriter, Vandhana Krishnan, Arash Alavi, Cuiping Pan.

**Funding acquisition:** Philip S. Tsao, Michael P. Snyder, Cuiping Pan.

**Investigation:** Amir Bahmani, Kyle Ferriter, Vandhana Krishnan, Arash Alavi, Amir Alavi, Cuiping Pan.

**Methodology:** Amir Bahmani, Kyle Ferriter, Cuiping Pan.

**Project administration:** Amir Bahmani, Michael P. Snyder, Cuiping Pan.

**Resources:** Amir Bahmani, Cuiping Pan.

**Software:** Amir Bahmani, Kyle Ferriter, Amir Alavi.

**Supervision:** Amir Bahmani, Philip S. Tsao, Michael P. Snyder, Cuiping Pan.

**Validation:** Amir Bahmani, Kyle Ferriter, Cuiping Pan.

**Visualization:** Amir Bahmani, Kyle Ferriter, Vandhana Krishnan, Arash Alavi, Cuiping Pan.

**Writing – original draft:** Amir Bahmani, Kyle Ferriter, Vandhana Krishnan, Arash Alavi, Amir Alavi, Philip S. Tsao, Michael P. Snyder, Cuiping Pan.

**Writing – review & editing:** Amir Bahmani, Kyle Ferriter, Vandhana Krishnan, Arash Alavi, Amir Alavi, Philip S. Tsao, Michael P. Snyder, Cuiping Pan.

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
