## [Decision Letter · Decision Letter 0]

3 Jan 2021

Dear Dr. Pan,

Thank you very much for submitting your manuscript "Swarm: A federated cloud framework for large-scale variant analysis" for consideration at PLOS Computational Biology.

As with all papers reviewed by the journal, your manuscript was reviewed by members of the editorial board and by several independent reviewers. In light of the reviews (below this email), we would like to invite the resubmission of a significantly-revised version that takes into account the reviewers' comments.

We cannot make any decision about publication until we have seen the revised manuscript and your response to the reviewers' comments. Your revised manuscript is also likely to be sent to reviewers for further evaluation.

Sincerely,

Mihaela Pertea

Software Editor

PLOS Computational Biology

Mihaela Pertea

Software Editor

PLOS Computational Biology

Reviewer's Responses to Questions

**Comments to the Authors:**

Reviewer #1: The authors presented a cloud computing framework for analyzing genomic variants across different cloud platforms. The proposed system, Swarm, is designed to minimize the need to move data across servers and expedite cross-analyses of genomic datasets reside in various cloud platforms. The authors conducted a series of evaluations using the data from the 1000 Genomes Project and various annotations, including dbNSFP 35a, ClinVar, gnomAD, Cosmic 70, and GTEx Analysis v7 eQTL. They showed that the proposed system achieved good performance in terms of computation time, the amount of data needed to be moved across platforms, and the computational cost on popular cloud computing platforms. Below are my comments.

General comments:

1. The variation in the execution time was not shown in Figures 2, 3, or 4. Although the variations may not be substantial since the pipeline under investigation is purely computational, as the authors stated on page 7, the information on the execution time variation can help determine whether one setting is statistically better than another. The authors could conduct significance tests to compare the differences across groups as well.

2. It was unclear how much time is needed to set up the Swarm environment proposed by the authors. The computational overhead of setting up the platform does not seem to be included in the current analyses. The authors could package the setup codes in a single executable or via docker to expedite and simplify the setup process.

3. Table 3 compares the average running time for querying an example single nucleotide polymorphism (SNP) using Apache Presto. The precision of the execution time is somewhat limited in the current table. The authors could consider using `time` or other related Unix command to get the exact running time of the query using different numbers of compute nodes. Metrics of variations in different runs will be helpful here as well.

4. The authors could discuss how their proposed framework could accommodate federated machine learning tasks. More and more users are developing machine learning approaches to aggregate the contributions of different genetic variants in relation to their outcomes of interest (such as diseases, phenotypes, or other endpoints). It would be interesting to see if the proposed system not only provides simple summary statistics or results from data queries but also enables the transfer of gradients or any other intermediates required for federated learning. This could greatly enhance the potential impact of the proposed cloud computing framework.

5. Figure 4 showed that the non-PVM (non-preemptible) environment and the N-2 PVMs (preemptible) + 2 non-PVMs setups have similar average execution time, while the monthly cost of the non-PVM environment is higher, since non-PVM generally cost more. Did the authors experience any preemption when running the experiments with PVM? If so, how does that affect the computation time and cost? What are the methods implemented in Swarm to enable a fast resumption of the unfinished computation?

6. The authors specified the compute nodes used for the third experiment (n1-standard-4 on Google Cloud Platform). However, the computing environment for other experiments was not specified. Different computational environments likely have different computational performance and cost.

7. Readers may be interested in a quick comparison between the proposed framework and some alternatives. For example, how is the computation time and cost of Swarm compare with a naïve implementation of SQL database that requires moving all relevant data across the platforms?

Additional minor comments:

1. Page 7, paragraph 4: “preemtible" should be “preemptible”.

2. For some reason, the GitHub Link provided in the manuscript does not work for me. I am not sure if it requires any specific permission setup (e.g., within the authors’ research groups) to access the source codes.

Reviewer #2: Bahmani et al. sescribe Swarm as a federated framework for variant analysis. Swarm offers to perform computational analyses on large genomics datasets hosted on different cloud platforms, enabling collaboration within or between different organizations and institutions, and facilitating

multi-cloud solutions. As such, the method is fairly generic and could accelerate discoveries in small teams and larger collaborative studies, including between research and healthcare. It could also reduce costs of scientific studies, given the data movement is expensive in the cloud world.

Frameworks for federated computation are likely to increase in importance, as are frameworks to

promote minimal data motion and facilitates crosstalk between datasets stored on different cloud platforms. This is an important contribution, and I overall liked this manuscript, although I have a couple of remaining questions:

The authors applied swarm to GCP BigQuery, AWS Athena and Apache Presto on an Apache Hadoop cluster. How easy is it to port swarm to new clouds, including non-commercial clouds and clouds used outside of the USA for example?

Could the authors present cost estimates for some generic operations? Specifically, what compute costs can be saved by using swarm when integrating data from distinct cloud platforms?

Additionally, the authors should discuss how anomalies are dealt with in the framework. In particular, is Swarm tolerant to issues/anomalies that may occur during cloud computing, and which unintendedly can increase the costs of a study quite substantially – for example by delaying the pace of a large-scale project (Yakneen et al. Nat Biotechnol 2020). Have the authors considered self-healing for Swarm?

I did not see a software maintenance plan. Will the swarm framework be actively maintained? This may be very important. Cloud frameworks are short lived, and this is a rapidly moving field.

Could swarm be used for functions beyond large-scale variant analysis?

**Have all data underlying the figures and results presented in the manuscript been provided?**

Reviewer #1: **No: **The GitHub Link provided in the manuscript does not work for me. I am not sure if it requires any specific permission setup (e.g., within the authors’ research groups) to access the source codes.

Reviewer #2: Yes

PLOS authors have the option to publish the peer review history of their article (what does this mean?). If published, this will include your full peer review and any attached files.

Reviewer #1: No

Reviewer #2: **Yes: **Jan Korbel
---

## [Decision Letter · Decision Letter 1]

18 Apr 2021

Dear Dr. Pan,

We are pleased to inform you that your manuscript 'Swarm: A federated cloud framework for large-scale variant analysis' has been provisionally accepted for publication in PLOS Computational Biology.

Best regards,

Mihaela Pertea

Software Editor

PLOS Computational Biology

Mihaela Pertea

Software Editor

PLOS Computational Biology

Reviewer's Responses to Questions

**Comments to the Authors:**

Reviewer #1: The authors have addressed my comments raised previously.

Reviewer #2: The authors have adequately responded to the requests made by the reviewers.

**Have the authors made all data and (if applicable) computational code underlying the findings in their manuscript fully available?**

Reviewer #1: Yes

Reviewer #2: Yes

PLOS authors have the option to publish the peer review history of their article (what does this mean?). If published, this will include your full peer review and any attached files.

Reviewer #1: No

Reviewer #2: No

---

## [Editor Report · Acceptance letter]

7 May 2021

PCOMPBIOL-D-20-02064R1 

Swarm: A federated cloud framework for large-scale variant analysis

Dear Dr Pan,

I am pleased to inform you that your manuscript has been formally accepted for publication in PLOS Computational Biology. Your manuscript is now with our production department and you will be notified of the publication date in due course.

With kind regards,

Katalin Szabo
